# Identification of Key Biomarkers Related to Lipid Metabolism in Acute Pancreatitis and Their Regulatory Mechanisms Based on Bioinformatics and Machine Learning

**DOI:** 10.3390/biomedicines13092132

**Published:** 2025-08-31

**Authors:** Liang Zhang, Yujie Jiang, Taojun Jin, Mingxian Zheng, Yixuan Yap, Xuanyang Min, Jiayue Chen, Lin Yuan, Feng He, Bingduo Zhou

**Affiliations:** 1Department of Gastroenterology, Yueyang Hospital of Integrated Traditional Chinese and Western Medicine, Shanghai University of Traditional Chinese Medicine, Shanghai 200437, China; 13391473771@163.com (L.Z.); yu990528@163.com (Y.J.); jintaojun96@163.com (T.J.); zmingxian1224@163.com (M.Z.); yapyixuan@163.com (Y.Y.); 13917099248@163.com (X.M.); 15901970941@163.com (J.C.); 2The Center for Cancer Research, School of Integrative Medicine, Shanghai University of Traditional Chinese Medicine, Shanghai 201203, China; yuanlin@shutcm.edu.cn

**Keywords:** acute pancreatitis, lipid metabolism, machine learning, bile acid metabolism, mitochondrial function

## Abstract

**Background:** Acute pancreatitis (AP) is characterized by the abnormal activation of pancreatic enzymes due to various causes, leading to local pancreatic inflammation. This can trigger systemic inflammatory response syndrome and multi-organ dysfunction. Hyperlipidemia, mainly resulting from lipid metabolism disorders and elevated triglyceride levels, is a major etiological factor in AP. This study aims to investigate the role of lipid metabolism-related genes in the pathogenesis of AP and to propose novel strategies for its prevention and treatment. **Methods:** We obtained AP-related datasets GSE3644, GSE65146, and GSE121038 from the GEO database. Differentially expressed genes (DEGs) were identified using DEG analysis and gene set enrichment analysis (GSEA). To identify core lipid metabolism genes in AP, we performed least absolute shrinkage and selection operator (LASSO) regression and support vector machine recursive feature elimination (SVM-RFE) analysis. Gene and protein interactions were predicted using GeneMANIA and AlphaFold. Finally, biomarker expression levels were quantified using Real-Time quantitative Polymerase Chain Reaction (RT-qPCR) in an AP mouse model. **Results:** Seven lipid metabolism-related genes were identified as key biomarkers in AP: *Amacr*, *Cyp39a1*, *Echs1*, *Gpd2*, *Osbpl9*, *Acsl4*, and *Mcee*. The biological roles of these genes mainly involve fatty acid metabolism, cholesterol metabolism, lipid transport across cellular membranes, and mitochondrial function. **Conclusions:** *Amacr*, *Cyp39a1*, *Echs1*, *Gpd2*, *Osbpl9*, *Acsl4*, and *Mcee* are characteristic biomarkers of lipid metabolism abnormalities in AP. These findings are crucial for a deeper understanding of lipid metabolism pathways in AP and for the early implementation of preventive clinical measures, such as the control of blood lipid levels.

## 1. Introduction

Acute pancreatitis (AP) is an inflammatory disease of the pancreas induced by various factors, which is mainly characterized by self-digestion of pancreatic tissue due to abnormal activation of pancreatic enzymes, local inflammatory reactions, and can even trigger systemic inflammatory response syndrome (SIRS) [1,2]. In severe cases, it can progress to multiple organ dysfunction syndrome (MODS) [3,4]. The etiology of AP is diverse, with major contributing factors including cholelithiasis, alcoholism, hyperlipidemia, and medications. Among these, hypertriglyceridemia (HTG) is a significant risk factor for AP [5]. In recent years, despite advances in diagnostic and therapeutic strategies for AP, the morbidity and mortality rates of severe acute pancreatitis (SAP) remain high [6]. This is particularly true for patients with concomitant lipid metabolism disorders, whose disease progresses more rapidly and who have a poorer prognosis [7]. Exploring the role of lipid metabolism in the development of AP and identifying potential molecular markers are crucial for early diagnosis, disease classification, and targeted treatment.

Lipid metabolism disorders, especially HTG, are significant risk factors for AP and may accelerate disease progression through multiple mechanisms [8]. Research has indicated that during AP, damaged adipocytes release pancreatic lipase, which then elevates triglyceride (TG) levels in pancreatic capillaries [9], leading to the release of free fatty acids (FFAs), which are categorized as saturated fatty acids (SFAs) and unsaturated fatty acids (UFAs). High concentrations of UFAs inhibit mitochondrial complexes in pancreatic alveolar cells, resulting in elevated intracellular Ca^2+^ concentrations, cytokine release, tissue damage, and reduced pancreatic duct function [10,11]. FFAs have activated inflammatory cytokines, including tumor necrosis factor (TNF)-α, Interleukin (IL)-6, IL-1β, and monocyte chemoattractant protein (MCP)-1, which exacerbate the inflammatory cascade in AP [12,13]. These findings suggest that lipid metabolism disorders are closely linked to the regulation of the local immune micro-environment of the pancreas. Abnormal expression of specific lipid metabolism-related genes may play a crucial role in AP progression. Notably, *ACSL4*, a gene involved in cell membrane lipid synthesis, has been shown to be central to AP pathology and may serve as a potential therapeutic target [14]. However, the molecular mechanisms by which lipid metabolism abnormalities regulate AP development remain unclear. A systematic analysis of the expression patterns of relevant genes and their regulatory mechanisms could enhance our understanding of AP pathogenesis and inform personalized treatment strategies.

Advancements in high-throughput sequencing and computational biology have made machine learning and bioinformatics essential tools for exploring disease diagnosis, treatment, and underlying pathological mechanisms. In this study, we conducted a systematic analysis of AP-related lipid metabolism core genes and their regulatory mechanisms by integrating gene expression data, gene enrichment analysis, machine learning, protein interaction networks, and metabolic pathway analysis [15,16,17,18]. We then experimentally validated the candidate genes using an AP mouse model to ensure the reliability and clinical translational value of the identified biomarkers.

This study aims to identify key lipid metabolism-related genes involved in the pathogenesis of acute pancreatitis and elucidate their core regulatory mechanisms through integrative bioinformatics, machine learning, and animal experiments.

## 2. Methods

### 2.1. Data Collection

Gene expression profiles were obtained from the Gene Expression Omnibus database (GEO, https://www.ncbi.nlm.nih.gov/geo/, accessed on 9 October 2024) [19] using the keyword “acute pancreatitis”. Three gene sets were selected: GSE3644, GSE65146, and GSE121038. GSE3644 contained 6 normal and 6 disease mice; GSE65146 included 9 normal and 5 disease mice; and GSE121038 included 7 normal and 8 disease mice (see Table 1). All three datasets met our stringent criteria and were characterized accordingly (see Appendix A).

### 2.2. Data Preprocessing

Gene expression profiles were obtained from the GEO database and processed to normalize inter-batch differences in the three gene sets using R software (version 4.4.1, https://www.r-project.org/, accessed on 9 October 2024). The raw data were first adjusted for batch effects using the sva R package 4.4.1 [20] to eliminate systematic biases from experimental batch or technical differences. Data aggregation was then assessed using boxplots and principal component analysis (PCA) to compare the data before and after normalization, ensuring consistency following batch effect correction.

### 2.3. DEGs Identification and Functional Enrichment Analysis

Differentially expressed genes (DEGs) were identified using the limma R package [21]. The normalized datasets of the three gene sets were then subjected to cluster expression analysis to screen for DEGs meeting the criteria of |log_2_FC| > 0.585 and adjusted *p*_FDR_ value (adj. *p*.Val) < 0.05. Volcano and heat maps were plotted using the R packages ggplot2 and pheatmap to visualize and analyze the DEGs. Gene Ontology (GO) and Kyoto Encyclopedia of Genes and Genomes (KEGG) enrichment analyses of the DEGs were performed using the clusterProfiler R package [22], and results were visualized with the enrichplot and ggplot2 R packages.

### 2.4. Screening of Lipid Metabolism Genes Strongly Associated with APs

Lipid metabolism-related genes with the stable identification number R-MMU-556833 were searched on the gene set enrichment analysis (GSEA) website (https://www.gsea-msigdb.org/gsea/index.jsp, accessed on 6 December 2024), which includes two gene sets, GSE25724 and GSE76896. DEG analysis was conducted using the R packages ggplot2 and pheatmap. Lipid metabolism genes were then intersected with AP-related DEGs using the VennDiagram R package to identify lipid metabolism genes involved in the diagnosis and progression of AP.

### 2.5. Further Screening of Genes Using Machine Learning Algorithms

After screening the common genes, the Least Absolute Shrinkage and Selection Operator (LASSO) regression algorithm was applied using the glmnet R package for regularization and feature selection to identify the most relevant genes and reduce overfitting [17,23]. Additionally, the support vector machine recursive feature elimination (SVM-RFE) algorithm was applied using e1071, kernlab, and caret R packages to further filter the most important genes for classification and optimize classification performance [24]. The genes identified by both algorithms were intersected to determine the genes specific to AP diagnosis related to lipid metabolism, using the VennDiagram R package. Box plots, created with the reshape2 and ggpubr R packages, were used to visualize the expression differences and significance of the selected genes across different groups. The correlation between the selected genes was analyzed using the PerformanceAnalytics R package.

### 2.6. Further Analysis of the Characterized Genes

To explore the interactions between these genes and their potential roles in biological processes, we predicted and validated their functional relationships through gene network analysis using GeneMANIA [25], revealing their cooperative roles in biological functions. Protein sequences of the gene-specific mouse proteins were obtained from NCBI (https://www.ncbi.nlm.nih.gov/, accessed 15 January 2025), and protein structure predictions were performed using AlphaFold (https://alphafoldserver.com/about, accessed 15 January 2025) [26], with a focus on their interconnections. Additionally, the selected genes were subjected to GSEA to investigate their roles in lipid metabolism-related functions and signaling pathways.

### 2.7. Establishment of an AP Mouse Model

C57BL/6J mice were purchased from Shanghai Slaughter Laboratory Animal Co., Ltd., Chinese Academy of Sciences(Shanghai, China), and had free access to sterile water and food. AP was induced using the retrograde biliopancreatic duct injection method. After anesthesia, the mice were fixed, and the abdomen was opened. The biliopancreatic duct was obstructed with an arterial clip near the hepatic hilar end, and a solution containing 2% sodium taurocholate and methylene blue was injected through a microsyringe pump at a rate of 0.01 mL/min for 5 min [27]. The abdomen was then sutured layer by layer. After surgery, the mice were placed on a heating pad, and their vital signs were monitored. They were transferred to cages upon awakening.

### 2.8. Pancreas Tissue Sampling and HE Staining

Histological analysis of tissues was performed as previously described [28,29]. Briefly, around 0.5 cm^3^ section of the pancreatic head was fixed in 4% paraformaldehyde for histopathological analysis. The remaining tissue was frozen in liquid nitrogen and stored at −80 °C for molecular biology analyses. The fixed pancreatic tissues were embedded in paraffin, and 4 μm-thick sections were prepared using a pathology slicer (Leica Biosystems, Shanghai, China). The sections were stained with hematoxylin and eosin (H&E) and examined using a digital slide scanning system (PreciPoint M8, Bad Homburg, Germany) following Schmidt’s pancreatic histopathology scoring criteria [30] (see Appendix A).

### 2.9. Real-Time Quantitative Polymerase Chain Reaction (RT-qPCR)

RT-qPCR was used to quantify the expression of genes in tissues, as previously described [31,32]. Briefly, RNA was extracted from pancreatic tissues using TRIzol, and reverse transcription was performed with the TAKARA RT Reagent Kit (TAKARA, RR047A, Beijing, China). RT-PCR was conducted using the TB Green qPCR Kit (TAKARA, RR420A, Beijing, China). The primer sequences are provided in Appendix A.

### 2.10. Data Processing

Data processing and graphing were performed using R (version 4.4.1). Student’s *t*-test was used for normally distributed data, the Wilcoxon rank-sum test for non-normally distributed data, and Spearman’s rank correlation test to assess the correlation between core genes and metabolic genes/pathways. One-way ANOVA followed by Dunnett’s multiple comparisons test was performed using GraphPad Prism version 10.0.0 for Windows (GraphPad Software, Boston, MA, USA, www.graphpad.com). Statistical significance was set at *p*_FDR_ < 0.05, with multiple comparisons corrected using the Benjamini–Hochberg method (FDR correction) (see Figure 1).

## 3. Result

### 3.1. Data Standardization and Elimination of Batch Differences

After correcting for batch effects in the three datasets, GSE3644, GSE65146, and GSE121038, obtained from the GEO database, boxplots revealed that the median expression values of the samples were well-converged (Figure 2A), indicating that systematic biases from different experimental platforms were effectively eliminated. PCA further confirmed the effectiveness of data integration. Prior to standardization, samples were significantly clustered by dataset source in PCA space. However, after correction, samples from the normal and AP groups were clearly separated, with inter-batch differences substantially reduced (Figure 2B), demonstrating that the data quality meets the requirements for subsequent analysis.

### 3.2. Screening of DEGs

Using the limma package, 1064 DEGs were identified, including 647 up-regulated genes and 417 down-regulated genes (adjusted *p*_FDR_ value < 0.05, |log_2_FC| > 0.585). Volcano plots displayed red for up-regulated genes, green for down-regulated genes, and gray for non-significant genes (Figure 2C). To further visualize the expression patterns of the top DEGs, heatmaps were generated, showing the top 50 up-regulated and 50 down-regulated DEGs. Red indicated up-regulated genes, and blue indicated down-regulated genes. Notably, lipid metabolism-related genes (e.g., Acsl4, Cyp39a1) were significantly up-regulated in the AP group, confirming that the expression patterns of DEGs clearly distinguished the normal group from the AP group (Figure 2D).

### 3.3. Gene Enrichment Analysis

To better understand the pathogenesis of AP, we conducted GO analysis and KEGG pathway enrichment analysis. GO analysis revealed that biological processes such as wound healing, cell adhesion, and RNA polymerase II-specific DNA binding were enriched in AP (Figure 2E). These processes are closely linked to intercellular interactions, tissue repair, and immune response, providing a biological basis for the connection between abnormal lipid metabolism and the inflammatory response. Subsequently, a ring diagram (Figure 2F) was used to illustrate the distribution and enrichment of the different gene sets in the GO analysis in terms of function.

Next, we conducted KEGG pathway enrichment analysis to further explore the biological significance of these genes. The results revealed several significantly enriched pathways related to AP, including the MAPK signaling pathway, the PI3K-Akt signaling pathway, and the pyroptosis and inflammatory response pathway, all of which are closely linked to immune response, lipid metabolism, and the pathomechanisms of AP. Additionally, KEGG analysis highlighted multiple metabolic pathways related to lipid metabolism, such as fatty acid and cholesterol metabolism. Aberrant regulation of these pathways may be closely associated with the onset and progression of AP (Figure 2G). The findings provide a detailed pathway-level analysis for understanding the interrelationship between immune response and lipid metabolism in AP.

### 3.4. Screening of AP-Related Lipid Metabolism Labeled Genes

We conducted differential expression analysis on lipid metabolism-related genes with the stable identification number R-MMU-556833 and identified a total of 625 key genes regulating lipid metabolism, which may serve as potential biomarkers for further investigation into the relationship between lipid metabolism and AP. The volcano plot displayed genes significantly up- and down-regulated in lipid metabolism (Figure 3A). The heatmap further visualized the top 50 up-regulated and top 50 down-regulated genes, with red indicating up-regulated genes and blue indicating down-regulated genes (Figure 3B). We performed an intersection analysis of lipid metabolism genes with previously identified AP-associated DEGs, revealing 54 overlapping genes (Figure 3C). These genes may play a crucial role in lipid metabolism abnormalities in AP.

### 3.5. Screening AP-Related Lipid Metabolism Signature Genes Using Machine Learning

To further identify core lipid metabolism genes associated with AP, we applied the LASSO regression and SVM-RFE algorithms. LASSO regression successfully identified eight core genes associated with AP, exhibiting an AUC value close to 1.0, indicating a very high diagnostic ability (Figure 3D). The SVM-RFE algorithm further optimized the feature set and identified 19 core genes (Figure 3E). The Venn diagram showed the intersection of genes selected by both LASSO regression and SVM-RFE (Figure 3F), revealing 7 common genes: *Amacr*, *Cyp39a1*, *Echs1*, *Gpd2*, *Osbpl9*, *Acsl4*, and *Mcee*.

### 3.6. Validation of Efficacy of Characterized Genes and Gene Interactions

To further analyze the seven core genes, we used box plots to visualize the expression differences across the two sample sets and assessed statistical significance, confirming their key roles in lipid metabolism abnormalities in AP (Figure 4A). Correlation analysis revealed potential interactions between these genes (Figure 4B), such as a strong positive correlation between *Osbpl9* and *Mcee* (r = 0.72, *p* < 0.001), suggesting a possible synergistic effect in AP. Additionally, a weak negative correlation was observed between *Cyp39a1* and *Echs1* (r = −0.41), which may indicate inhibitory interactions in certain signaling pathways. The gene interaction network constructed by GeneMANIA revealed that the core genes jointly regulate fatty acid metabolism, cholesterol metabolism, and energy metabolism through close interactions. Additionally, these genes may be linked to inflammatory and metabolic stress responses, as seen with genes such as *Sirt3*, *Clybl*, and *Aldh6a1*. Interactions with other metabolism-related genes suggest their role in maintaining metabolic homeostasis and mitigating inflammation. *Amacr* and *Cyp39a1* are involved in bile acid metabolism through co-expression and physical interactions, while *Acsl4* forms a functional complex with *Echs1* in the mitochondrial β-oxidation pathway (Figure 4C).

### 3.7. GSEA Enrichment Analysis of Characterized Genes

We performed GSEA on the seven identified genes to analyze their functions and associated pathways. *Amacr*-related genes were enriched in peroxisomal fatty acid β-oxidation and bile acid biosynthesis (Figure 5(A1,A2)), and KEGG analysis confirmed their strong links to primary bile acid synthesis and peroxisomal pathways (Figure 5(A3,A4)), suggesting that *Amacr* contributes to AP pathology through bile acid metabolism abnormality. *Cyp39a1*, involved in cholesterol hydroxylation and oxysterol metabolism (Figure 5(B1,B2)), showed significant correlations with cholesterol metabolism and the NF-κB signaling pathway in KEGG analysis (Figure 5(B3,B4)), indicating that its dysregulation may exacerbate pancreatic injury by amplifying inflammatory signals. *Echs1*-related genes were enriched in FAO and acyl-CoA hydratase activity (Figure 5(C1,C2)), with KEGG analysis linking them to fatty acid degradation and oxidative phosphorylation (Figure 5(C3,C4)), suggesting that disturbed mitochondrial energy metabolism drives AP progression. Gpd2, involved in glycerophosphate shuttling and NADH oxidation (Figure 5(D1,D2)), correlated with glycolysis and insulin resistance pathways (Figure 5(D3,D4)), suggesting its role in aggravating AP through energy metabolism reprogramming. Osbpl9-related genes were enriched in cholesterol transport and lipid raft assembly (Figure 5(E1,E2)), and KEGG analysis showed significant correlations with atherosclerosis and toll-like receptor (TLR) signaling pathways (Figure 5(E3,E4)), indicating that *Osbpl9* may drive AP via lipid-inflammation interactions. *Acsl4*, involved in long-chain fatty acid activation and ferroptosis regulation (Figure 5(F1,F2)), was linked to ferroptosis and arachidonic acid metabolism (Figure 5(F3,F4)), suggesting it mediates cell death through lipid peroxidation. Mcee-related genes were enriched in branched-chain amino acid catabolism and methylmalonyl-CoA metabolism (Figure 5(G1,G2)). KEGG analysis showed links to propionate metabolism and oxidative stress (Figure 5(G3,G4)), suggesting that Mcee downregulation may exacerbate AP through toxic metabolite accumulation.

### 3.8. Protein Structure Prediction and Interaction Analysis of Characterized Genes

We used AlphaFold to predict the three-dimensional (3D) structures and protein interactions of proteins encoded by the seven core genes. Since FASTA sequences for Gpd2 and Mcee were unavailable for the mouse host, 3D models were generated only for the remaining five genes. Subsequently, we predicted the protein–protein interactions among these five core genes (see Appendix A). The predicted structures of *Amacr*, *Cyp39a1*, *Echs1*, and *Acsl4* pTM values of 0.94, 0.86, 0.95, and 0.92, respectively—all above 0.8—indicating high accuracy. Osbpl9 showed a pTM value of 0.67, consistent with high-quality modeling. Predicted interaction structures between Amacr&Acsl4 and Echs1&Acsl4 had pTM values of 0.62 and 0.7, respectively, suggesting reliable inter-protein modeling. Additional interations-Amacr&Echs1 (0.57), Amacr&Osbpl9 (0.55), Cyp39a1&Acsl4 (0.56), Echs1&Osbpl9 (0.51), and Osbpl9&Acsl4 (0.52)—produced moderate pTM values, indicating structural predictions similar to native conformations. In contrast, predictions for Amacr&Cyp39a1 (0.46), Cyp39a1& Echs1 (0.43), and Cyp39a1&Osbpl9 (0.43) yielded lower pTM values, suggesting limited confidence in these interaction models.

### 3.9. Animal Experimental Validation of Core Genes

To verify the expression levels of lipid metabolism-related genes in AP, we collected pancreatic tissues from 15 AP model mice and 15 healthy mice. Among them, 7 were randomly selected for paraffin embedding and sectioning for HE staining in each group, while the other 8 were used for RNA extraction and RT-qPCR analysis. In RT-qPCR experiments, Actb was used for the internal control gene. Pancreatic tissue injury was evaluated by histological examination and pathology scoring. In the control group, pancreatic architecture was preserved, with intact acinar structure, clearly defined lobules, and no evident congestion or edema. In contrast, the AP group exhibited disintegration and disruption of pancreatic acinar structures, interstitial edema with hemorrhagic foci, loss of lobular architecture, marked inflammatory cell infiltration, and extensive coagulative necrosis. Pathology scores were significantly higher in the AP group than in controls (*p* < 0.0001), confirming the success and reliability of the animal model. RT-qPCR results showed significant upregulation of *Amacr*, *Cyp39a1*, *Echs1*, and *Mcee*, and significant downregulation of *Gpd2*, *Osbpl9*, and *Acsl4* in AP mice, consistent with the gene expression patterns identified through machine learning validation (Figure 6).

## 4. Discussion

In this study, seven core genes (*Amacr*, *Cyp39a1*, *Echs1*, *Gpd2*, *Osbpl9*, *Acsl4*, and *Mcee*) closely associated with lipid metabolism disorders in AP were systematically identified through the integration of bioinformatics and machine learning approaches. These genes were found to contribute to AP progression by modulating fatty acid oxidation, cholesterol metabolism, mitochondrial function, and inflammatory pathways. Animal experiments further validated these findings. RT-qPCR analysis showed significant downregulation of *Amacr*, *Cyp39a1*, *Echs1*, and *Mcee*, and significant upregulation of *Gpd2*, *Osbpl9*, and *Acsl4* in AP model mice, consistent with the predictions reported by Sangshin.

Disorders of lipid metabolism are well-established as key etiological factors in AP [5,8,33]. The underlying mechanisms primarily involve adipose tissue inflammation, oxidative stress in pancreatic cells, and pancreatic tissue injury induced by imbalances in triglyceride, fatty acid, and cholesterol metabolism [34,35,36,37,38]. Numerous studies have demonstrated that the seven core genes identified in this study play critical roles in regulating lipid metabolism processes. *Amacr* regulates the peroxisomal degradation of branched-chain fatty acids [39,40]. *Cyp39a1* converts inflammatory hydroxysterols, such as 24-hydroxycholesterol (HC), into less active 7-hydroxysterols, thereby maintaining the metabolic balance of neurosterols and bile acids [41,42,43]. *Echs1* is a key enzyme in the mitochondrial fatty acid β-oxidation (FAO) pathway, supplying acetyl coenzyme A for the tricarboxylic acid (TCA) cycle [44,45]. *Gpd2* catalyzes the oxidation of glycerol-3-phosphate (G3P) to dihydroxyacetone phosphate (DHAP) [46], with G3P serving as a precursor for triglyceride and phospholipid synthesis [47]. *Osbpl9*, a member of the oxysterol-binding protein (*OSBPL*) family, binds oxidized cholesterol derivatives such as 25-HC and plays a central role in intracellular cholesterol and phospholipid transport [48]. *Acsl4* is critical in FA metabolism, facilitating the esterification of arachidonic acid with CoA and catalyzing lipid peroxidation, thereby promoting ferroptosis, autophagy, and apoptosis [49,50,51]. *Mcee*, a key enzyme in branched-chain fatty acid metabolism, provides essential intermediates for the TCA cycle [52].

The integration of bioinformatic analysis and experimental validation represents a major strength of this study. Using LASSO combined with the SVM-RFE algorithm, we further screened lipid metabolism-related genes with specificity to AP and identified seven core genes with significantly altered expression in the AP mouse model. GeneMANIA network analysis further demonstrated that these genes form functional modules through co-expression or physical interactions. For example, Acsl4 and *Echs1* appear to synergistically regulate energy metabolism within the mitochondrial β-oxidation pathway. In this study, animal experiments not only confirmed pancreatic tissue injury through HE staining but also quantified gene expression changes using RT-qPCR, thereby validating the reliability of the bioinformatic predictions. This multidimensional analysis strategy offers a novel methodological framework for investigating the molecular mechanisms of AP. Unlike previous studies, the novelty of this work lies in the first systematic integration of lipid metabolism-related genes in the context of AP and the identification of a synergistic multigene regulatory network underlying disease progression. Although *Amacr* is commonly regarded as a biomarker for prostate cancer [53,54], we found it to be significantly downregulated in AP. This downregulation may cause abnormal accumulation of toxic C27-cholestate intermediates, which can inhibit mitochondrial complexes I and IV, reduce ATP production, elevate cellular reactive oxygen species levels, and ultimately trigger apoptosis [55,56]. *Cyp39a1*, typically recognized as a specific marker of hepatocellular carcinoma, was also found to be downregulated in AP [57]. This reduction may lead to aberrant accumulation of inflammatory hydroxysterols, such as 24-HC, which activates the NLRP3 inflammasome, promotes maturation and secretion of IL-1β and IL-18, and contributes to the formation of a localized inflammatory microenvironment in the pancreas [58]. Dysregulation of *Cyp39a1* may also influence bile acid composition, thereby altering intestinal microbial metabolites and modulating the intestinal-pancreatic axis inflammatory response. This disruption can contribute to changes in the pancreatic inflammatory microenvironment, ultimately leading to lipid metabolism disorders and enhanced inflammatory and immune responses in diseases such as AP [59]. *Echs1* plays a critical role in mitochondrial energy conversion. In AP, disruption of *Echs1* activity leads to the accumulation of FAs and toxic lipid intermediates [60,61], as well as combined defects in FAO and OXPHOS [62]. These impairments result in reduced mitochondrial membrane potential and elevated ROS levels, disrupting cellular energy homeostasis, exacerbating oxidative stress, and aggravating AP pathology. *Gpd2* is closely associated with macrophage-mediated inflammatory responses and glycolysis in cancer cells [63,64]. Its upregulation in AP may represent an adaptive cellular response to metabolic stress. However, this alteration can disrupt the mitochondrial redox state, activate inflammatory signaling pathways, and exacerbate pancreatic injury. Additionally, enhanced *Gpd2* activity may lead to excessive consumption of G3P, resulting in imbalanced lipid metabolism and abnormal lipid accumulation [65]. *Osbpl9* has been relatively understudied and previously linked primarily to the tumor microenvironment [66,67]. In this study, we demonstrate for the first time that Osbpl9 may exacerbate AP by promoting de novo lipogenesis and inducing abnormal intracellular triglyceride accumulation in pancreatic cells, thereby intensifying lipid–inflammatory interactions [68]. *Acsl4* has been recognized as a key regulator of ferroptosis. Findings from this study suggest that *Acsl4* is not only a central driver of lipid metabolism disorders in AP but also a critical molecular link between metabolic dysregulation and inflammatory cascades. Its upregulation in AP promotes the accumulation of lipid peroxidation products, induces ferroptotic death of pancreatic acinar cells, and activates the TLR4/nuclear factor kappa B (NF-κB) signaling pathway. This activation leads to the upregulation of chemokines such as IL-1β and C-X-C motif chemokine ligand (CXCL)1, recruitment of neutrophils, amplification of local inflammation [69], and promotion of macrophage M1 polarization [70]. In addition, *Acsl4* may impair the clearance of damaged organelles by inhibiting autophagic lysosomal function, resulting in the intracellular accumulation of toxic lipids [71,72]. Defective *Mcee* function is commonly observed in mutational disorders [73]; however, this study is the first to propose that its downregulation in AP leads to the accumulation of methylmalonyl-CoA, which is converted into methylmalonic acid. This metabolite inhibits mitochondrial complexes I and II, induces oxidative stress, and promotes the release of inflammatory mediators [74,75]. These findings not only enhance our understanding of the pathological mechanisms underlying AP but also identify potential targets for precise therapeutic intervention.

AP is associated with poor clinical outcomes, and early diagnosis is essential for improving prognosis [76,77]. These characterized genes, especially *Acsl4*, exhibit high diagnostic efficacy and may serve as valuable biomarkers for clinical testing. These findings suggest the promising potential for the development of rapid diagnostic kits based on multigene panels. This approach could not only facilitate early identification of patients at high risk of progressing to SAP but also inform personalized therapeutic strategies. For example, patients with high *Acsl4* expression may benefit from targeted interventions such as antioxidants (e.g., N-acetylcysteine) [78] or ferroptosis inhibitors like Ferrostatin-1 [79], aimed at blocking lipid peroxidation cascade reactions. In the context of drug development, targeted therapeutic strategies can be designed to address the acute pathological processes of AP. For example, small-molecule inhibitors of *Acsl4* could be developed to block lipid peroxidation in pancreatic acinar cells. *Cyp39a1* agonists may help restore the pancreatic microenvironment; protective agents targeting *Echs1* and *Gpd2* could mitigate oxidative stress resulting from mitochondrial energy crises, and personalized dietary interventions based on *Osbpl9* and *Mcee* expression levels may help correct metabolic imbalances [80,81].

In conclusion, this study elucidates the core regulatory network underlying lipid metabolism disorders in AP through a multidisciplinary integrative approach, providing a foundation for further exploration of its molecular mechanisms and the development of novel diagnostic and therapeutic strategies. However, several limitations should be acknowledged. First, the bioinformatic analysis relied on publicly available databases, which may introduce selection bias. Second, the dataset used for computational analysis was limited in scope, comprising only animal-derived samples, potentially reducing its translational relevance. Additionally, the sample size for animal experiments was relatively small, and the experimental methods employed were basic, lacking validation using clinical tissue samples. Finally, the regulatory mechanisms of some identified genes remain incompletely elucidated and require further investigation. Future research should expand the sample cohort and integrate single-cell sequencing with multi-omics approaches to achieve a more comprehensive understanding of gene functions. The pathological roles of key genes ought to be further validated using gene editing models. These efforts, combined with strategies for early prevention and timely intervention, are expected to provide valuable insights into the prevention, diagnosis, and treatment of clinical AP, thereby facilitating the translation of basic research into clinical practice.

## 5. Conclusions

Based on bioinformatics analysis and machine learning, we systematically identified seven signature genes-*Amacr*, *Cyp39a1*, *Echs1*, *Gpd2*, *Osbpl9*, *Acsl4*, and *Mcee*-that contribute to oxidative stress and immune dysregulation through disturbances in fatty acid oxidation, cholesterol homeostasis, mitochondrial function, and inflammatory signaling. These alterations play a critical role in the progression of AP. This study not only advances understanding of the molecular link between AP and lipid metabolism disorders but also provides a novel framework for exploring early prevention and therapeutic intervention strategies in AP.

## Figures and Tables

**Figure 1 biomedicines-13-02132-f001:**
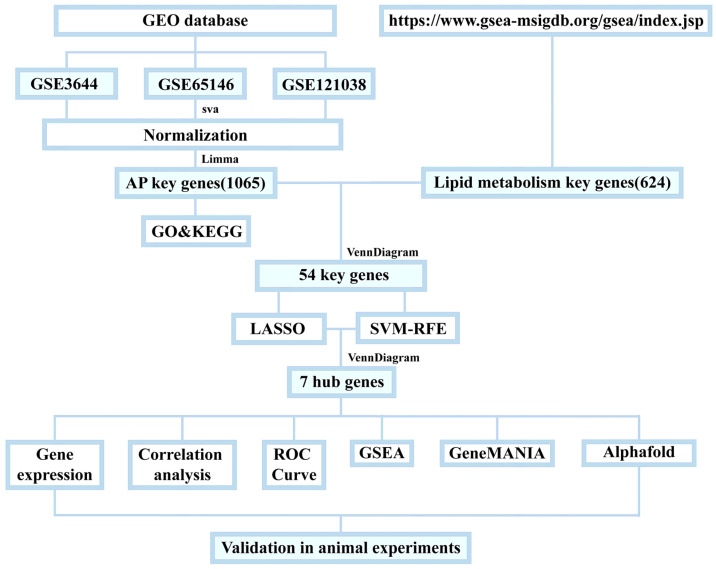
Flowchart of analytical steps in this study.

**Figure 2 biomedicines-13-02132-f002:**
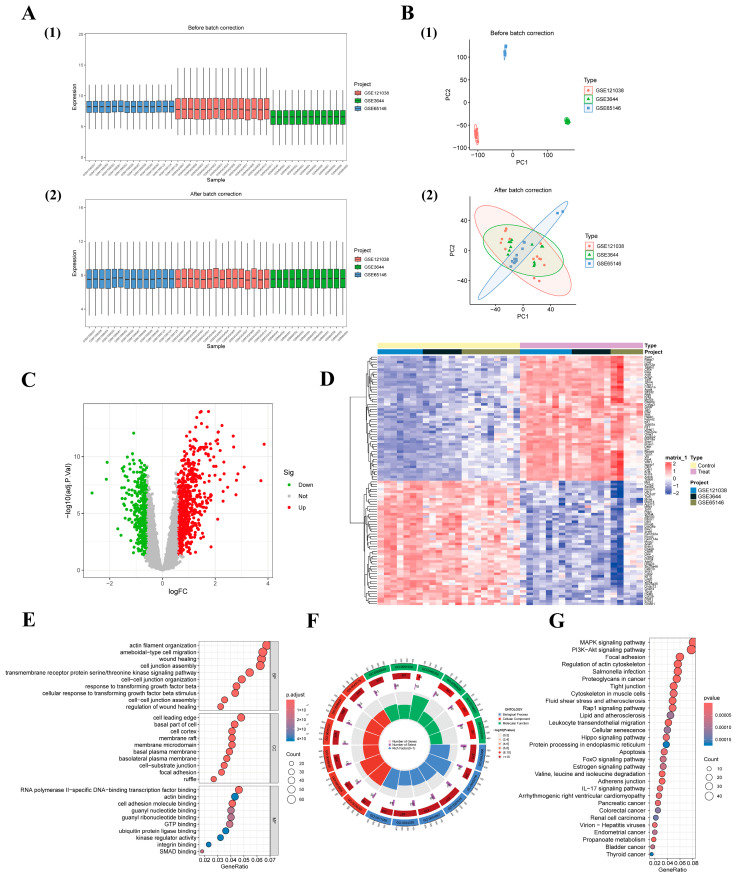
Pre-processing and analysis of raw data (**A**) boxplot of sample expression values before and after normalization (**B**) PCA demonstrating the distribution of samples before and after batch correction (**C**) volcano plot demonstrating the DEG of AP mice relative to healthy control mice (**D**) heatmap demonstrating the expression of the TOP50 differentially expressed genes (**E**) Results of the GO analysis demonstrating a significant enrichment of the AP in the biological processes, cellular components, and molecular functions (**F**) GO analysis loops demonstrating the distribution of lipid metabolism-related gene sets in various functional annotations (**G**) Details of KEGG enrichment analysis demonstrating pathway enrichment for lipid metabolism and immune responses in AP.

**Figure 3 biomedicines-13-02132-f003:**
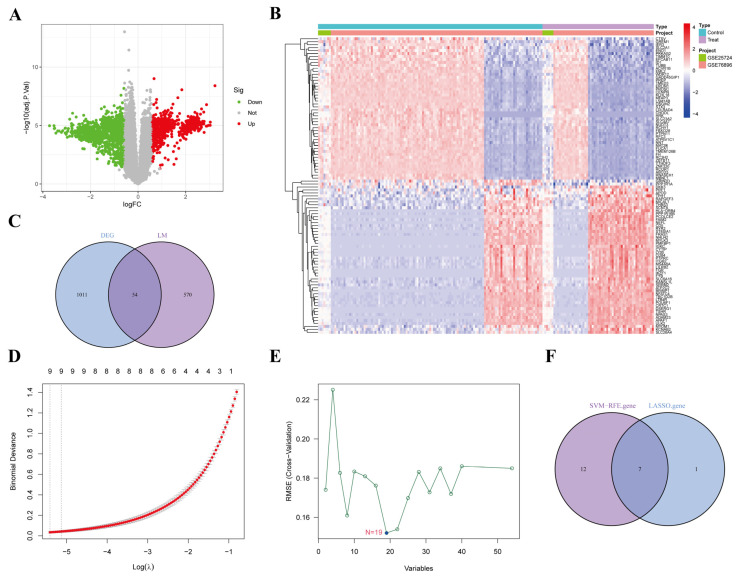
Screening of differentially expressed genes for AP-associated lipid metabolism (**A**) Volcano plot showing the results of differential expression analysis of the lipid metabolism gene set (R-MMU-556833) (**B**) Heatmap of TOP50 lipid metabolism genes (**C**) Venn diagram of lipid metabolism genes with AP-associated DEGs, revealing 54 common differentially expressed genes as potential biomarkers for AP markers. (**D**) Binomial Deviance variation in LASSO regression, demonstrating the effect of different regularization parameters on model performance. (**E**) SVM-RFE analysis demonstrating the process of feature selection. (**F**) Venn diagram of 7 common genes screened by LASSO regression with SVM-RFE.

**Figure 4 biomedicines-13-02132-f004:**
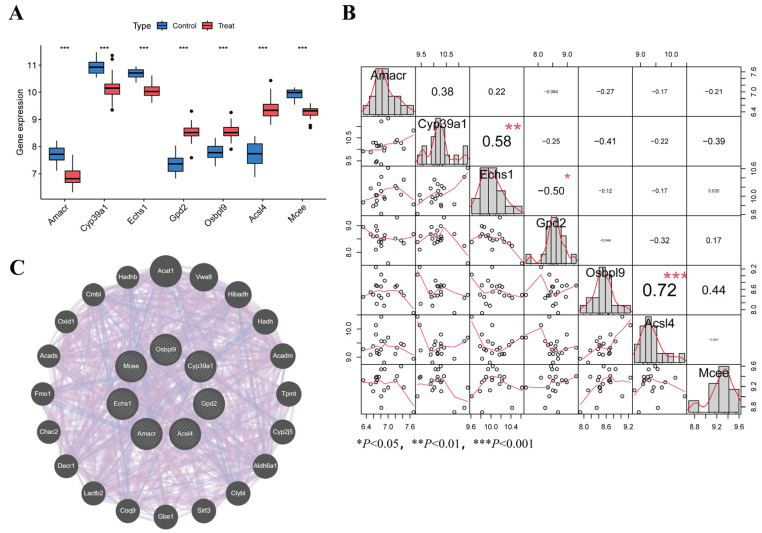
AP-associated lipid metabolism signature gene analysis and validation (**A**) Visualization of differential expression of differentially expressed genes in different groups (**B**) Correlation analysis plot (**C**) Characterized gene interaction network constructed by GeneMANIA.

**Figure 5 biomedicines-13-02132-f005:**
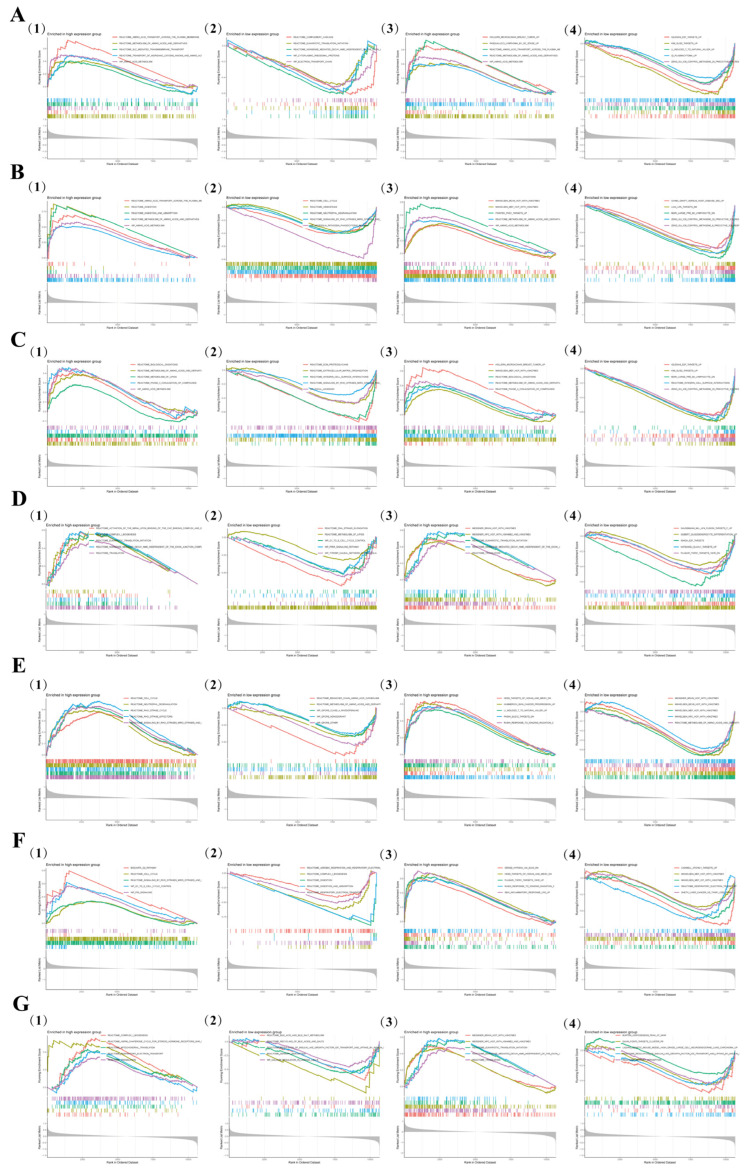
(**A**–**G**) GO and KEGG enrichment results of *Amacr*, *Cyp39a1*, *Echs1*, *Gpd2*, *Osbpl9*, *Acsl4*, and *Mcee*: (**1**,**2**) GO analyses demonstrating biological processes or molecular functions in which the top two were significantly enriched; (**3**,**4**) KEGG analyses demonstrating relevant metabolic or signaling pathways.

**Figure 6 biomedicines-13-02132-f006:**
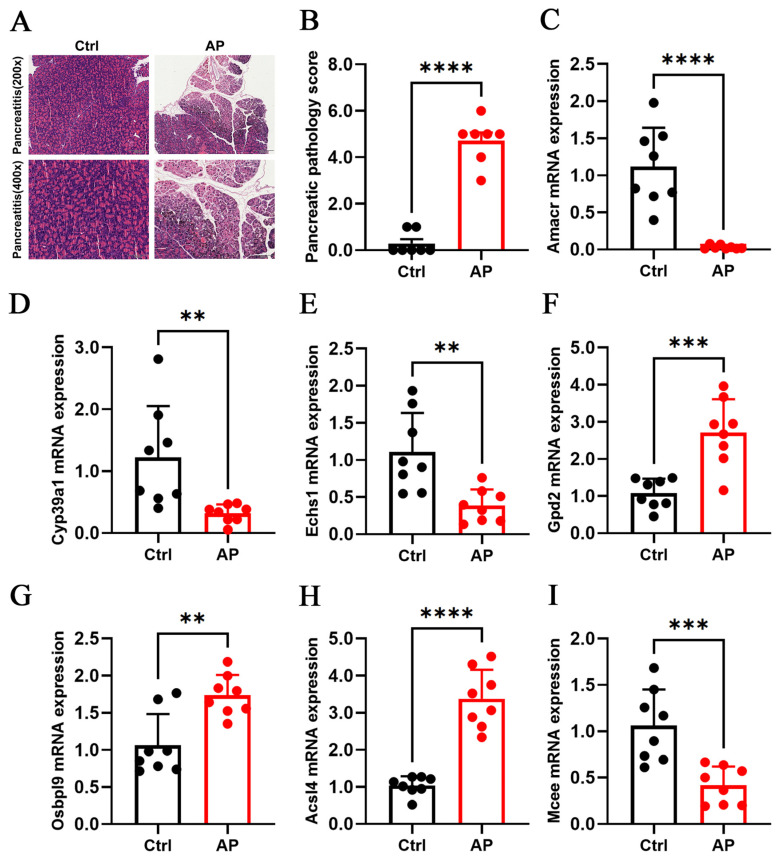
Validation of core genes at the animal level (**A**) HE staining of pancreatic tissues (n = 7) (**B**) Pathological scoring of pancreatic tissues (n = 7), **** *p* < 0.0001 (**C**–**I**) qPCR for *Amacr*, *Cyp39a1*, *Echs1*, *Gpd2*, *Osbpl9*, *Acsl4*, and *Mcee* expression (n = 8), ** *p* < 0.01, *** *p* < 0.001, **** *p* < 0.0001.

**Table 1 biomedicines-13-02132-t001:** Basic information on datasets used in the study.

Datasets	Type	Sample Size		Platforms
		Normal	Acute Pancreatitis	
GSE3644	RNA	GSM84549 GSM84550 GSM84551 GSM84555 GSM84556 GSM84557	GSM84552 GSM84553 GSM84554 GSM84558 GSM84559 GSM84560	GPL339
GSE65146	RNA	GSM1588057 GSM1588058 GSM1588059 GSM1588060 GSM1588086 GSM1588087 GSM1588088 GSM1588089 GSM1588090	GSM1588081 GSM1588082 GSM1588123 GSM1588124 GSM1588125	GPL6246
GSE121038	RNA	GSM3424897 GSM3424898 GSM3424899 GSM3424904 GSM3424905 GSM3424906 GSM3424907	GSM3424900 GSM3424901 GSM3424902 GSM3424903 GSM3424908 GSM3424909 GSM3424910 GSM3424911	GPL10787

## Data Availability

The datasets generated and/or analyzed during this study are available at Gene Expression Omnibus datasets (GEO, https://www.ncbi.nlm.nih.gov/geo/, accessed on 9 October 2024 GSE3644, GSE65146, GSE121038) and https://www.gsea-msigdb.org/gsea/index.jsp, accessed on 16 November 2024, were obtained and analyzed in Gene Set Enrichment Analysis (GSEA, http://www.gsea-msigdb.org/, accessed on 6 December 2024), Gene Ontology (GO, https://geneontology.org/, accessed on 24 December 2024), and Kyoto Encyclopedia of Genes and Genomes (KEGG, https://www.genome.jp/kegg/, accessed on 24 December 2024), and further inquiries can be directed to the corresponding author.

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
