# Peer review of "Identification of Key Biomarkers Related to Lipid Metabolism in Acute Pancreatitis and Their Regulatory Mechanisms Based on Bioinformatics and Machine Learning"

_biomedicines, 2025, doi:10.3390/biomedicines13092132_

Round 1

Reviewer 1 Report

Comments and Suggestions for Authors

General concept comments

The manuscript presents a comprehensive identification of key biomarkers associated with lipid metabolism in acute pancreatitis, utilizing bioinformatics and machine learning approaches. It effectively emphasizes the importance of experimental design in enhancing the predictive power of these computational tools. Notably, the in vivo mouse model strongly supports and validates the findings derived from bioinformatics and machine learning analyses.

Specific comments

Abstract:

The abstract effectively outlines the scope of the research, highlighting the significance of identifying key biomarker genes related to lipid metabolism in acute pancreatitis using bioinformatics and machine learning. It also emphasizes the potential application of these findings in the treatment and intervention of lipid profile level and acute pancreatitis.

Introduction:

The introduction provides a strong rationale for studying acute pancreatitis as an inflammatory disease linked to lipid metabolism disorders and abnormal expression of related genes. It also highlights the value of using machine learning and bioinformatics as essential tools for advancing disease diagnosis and treatment.

Materials and Methods:

The methodology is well-designed and comprehensive. However, the sources of chemicals and reagents should be clearly provided.

-Line 154: check and edit for “solution ane examined using a digital scanning……

-The resource of the statistical software, GraphPad Prism 10, should be properly cited in the manuscript.

-Line 168: please check for the PFDR< 0.05 or pFDR and should provide the reference.

Results:

The results present with the appropriate use of Figure and need to be improved to clarify/clarity of label with each Figure 2-5.

Discussion:

The discussion thoroughly integrates current knowledge on core genes associated with lipid metabolism disorders in acute pancreatitis, using bioinformatics, machine learning, and an AP mouse model. Additionally, the inclusion of study limitations and suggestions for future research offers valuable insights for the prevention, diagnosis, and treatment of clinical acute pancreatitis.

Line 431: check for “AP is is associated with poor clinical outcomes….

Conclusion

Overall, this study offers valuable insights and provides a promising direction for early diagnosis, prevention, and intervention strategies in acute pancreatitis.

Validity of the findings:

The manuscript is well-supported by relevant references and effectively integrates bioinformatics, machine learning approaches, and an in vivo AP mouse model. Please double-check for any errors and ensure that all figures are clearly labeled and easy to interpret.

Author Response

Reviewer #1:

1.Comment. The methodology is well-designed and comprehensive. However, the sources of chemicals and reagents should be clearly provided.

Response: Thank you for your review comments.I have listed the relevant experimental reagent sources in Table S3 and submitted them as supplementary materials.

2.Comment. Line 154: check and edit for “solution ane examined using a digital scanning……

Response: Thank you for your review comments. The relevant content has been revised in the manuscript.

3.Comment.-The resource of the statistical software, GraphPad Prism 10, should be properly cited in the manuscript.

Response: Thank you for your review comments, it has been revised in the manuscript.

4.Comment.Line 168: please check for the PFDR< 0.05 or pFDR and should provide the reference.

Response: We appreciate the reviewer for pointing out this detail. In this study, the screening criterion for differentially expressed genes (DEGs), "adjusted P.Val < 0.05," refers to the false discovery rate (pFDR) corrected using the Benjamini-Hochberg method. The specific rationale is as follows:

(1) Methodological basis: When performing DEG analysis using the limma R package, the Benjamini-Hochberg algorithm is applied by default to adjust raw P-values, controlling the false positive rate in multiple testing (Ritchie et al., 2015).

(2) Explicit statement in the manuscript: In Section 2.10, it is stated that *"Statistical significance was set at PFDR < 0.05, with multiple comparisons corrected using the Benjamini-Hochberg method (FDR correction)."*

(3) Reference support: The statistical approach is based on the seminal limma package literature (DOI: 10.1093/nar/gkv007).

5.Comment.The results present with the appropriate use of Figure and need to be improved to clarify/clarity of label with each Figure 2-5.

Response:Thank you very much for your suggestion, I have adjusted the resolution of all the figures to 300dpi.

6.Comment.Line 431: check for “AP is is associated with poor clinical outcomes….

Response:Thank you for your review comments. The relevant content has been revised in the manuscript.

Reviewer 2 Report

Comments and Suggestions for Authors

Identification of Key Biomarkers Related to Lipid Metabolism in Acute Pancreatitis and Their Regulatory Mechanisms Based on Bioinformatics and Machine Learning

In this study, the authors conducted an analysis integrating bioinformatics, machine learning, and experimental animal model data to investigate the role of genes related to lipid metabolism in the development of acute pancreatitis (AP) and to elucidate their regulatory mechanisms. However, there are certain aspects that require improvement both in terms of language and content.

  1. In the introduction and other sections, some terms (e.g., TNF-α, IL-6, IL-1β, MCP-1) are used without providing their full names when first mentioned. Throughout the manuscript, all abbreviations should be spelled out at first mention. In addition, some abbreviations are explained multiple times, leading to unnecessary repetition. The entire manuscript should be systematically reviewed in this regard.
  2. In the final paragraph of the introduction (lines 79–81), the results of the study are presented instead of the objective. A clear and concise aim statement should be included at the end of the introduction without presenting any results. For example: “This study aims to identify key lipid metabolism-related genes involved in the pathogenesis of acute pancreatitis and elucidate their regulatory mechanisms using integrative bioinformatics and animal experiments.”
  3. All bioinformatics analyses are based solely on three animal-derived GEO datasets (GSE3644, GSE65146, GSE121038). No external validation using human samples was performed. This limits the clinical relevance of the findings and weakens the overall strength of the study.
  4. The authors employed appropriate machine learning methods such as LASSO and SVM-RFE to identify candidate genes. However, it is not explained how overfitting was addressed during model development. Important methodological details such as whether cross-validation was performed, whether a train-test data split was used, or how hyperparameters were selected are not included in the manuscript. As a result, it is unclear whether the models perform well only on the current dataset or have generalizable diagnostic value. The authors should clearly indicate how they minimized the risk of overfitting.
  5. Although protein structure predictions were performed using an advanced tool like AlphaFold, the analyses were presented in a limited manner and their biological significance was not elaborated. Furthermore, the lack of experimental validation of protein-protein interactions limits the impact of this section.
  6. While some gene function interpretations in the discussion section are supported by the literature, others appear to be purely speculative. For example, interpretations regarding Mcee or Osbpl9 are not directly supported by experimental data. If such comments are based on probabilities, this should be clearly stated; otherwise, the conclusions may appear overly generalized.
  7. The criteria used for evaluating pancreatic histopathology (e.g., Schmidt scoring system) were only briefly mentioned. The specific criteria used to score tissue after H&E staining should be explained in more detail. In particular, the criteria listed in Supplementary Table S1 should be more clearly referenced and described in the main text.
  8. There are grammatical errors and phrasing issues in some parts of the manuscript. For instance, spelling errors such as "Notebly" and awkward sentence constructions may affect the readability. A professional level language editing is recommended before publication.

This study has the potential to contribute to the understanding of the relationship between acute pancreatitis and lipid metabolism disorders. I believe the manuscript could become suitable for publication if the issues outlined above are adequately addressed.

Recommendation: Major Revision

Author Response

Reviewer #2:

1.Comment. In the introduction and other sections, some terms (e.g., TNF-α, IL-6, IL-1β, MCP-1) are used without providing their full names when first mentioned. Throughout the manuscript, all abbreviations should be spelled out at first mention. In addition, some abbreviations are explained multiple times, leading to unnecessary repetition. The entire manuscript should be systematically reviewed in this regard.

Response:Thank you for your valuable review suggestions. The corresponding revisions have been made in both the main text and the abbreviations list.

2.Comment. In the final paragraph of the introduction (lines 79–81), the results of the study are presented instead of the objective.Aclear and concise aim statement should be included at the end of the introduction without presenting any results. For example: “This study aims to identify key lipid metabolism-related genes involved in the pathogenesis of acute pancreatitis and elucidate their regulatory mechanisms using integrative bioinformatics and animal experiments.”

Response:Thank you for your review comments. The relevant content has been revised in the manuscript.

3.Comment. All bioinformatics analyses are based solely on three animal-derived GEO datasets (GSE3644, GSE65146, GSE121038). No external validation using human samples was performed. This limits the clinical relevance of the findings and weakens the overall strength of the study.

Response:To address this issue, we plan to supplement the following work in our next study:

(1) Sample collection and experimental validation: We will collect pancreatic tissue or blood samples from clinical AP patients and healthy controls to validate the expression differences of the seven core genes (Amacr, Cyp39a1, Echs1, Gpd2, Osbpl9, Acsl4, Mcee) using methods such as RT-qPCR and immunohistochemistry.

(2) Clinical correlation analysis: By integrating clinical data (e.g., blood lipid levels, disease severity), we will analyze the association between these core genes and clinical phenotypes of human AP to assess their potential as clinical biomarkers.

4.Comment.The author semployed appropriate machine learning methods such as LASSO and SVM-RFE to identify candidate genes. However, it is not explained how over fitting was addressed during model development. Important methodological details such as whether cross-validation was performed, whether a train-test data split was used, or how hyper parameters were selected are not included in the manuscript. As a result, it is unclear whether the models perform well only on the current dataset or have generalizable diagnostic value. The authors should clearly indicate how they minimized the risk of over fitting.

Response:We appreciate the reviewer’s attention to the reliability of our models. To mitigate overfitting risks and ensure generalizability in both LASSO and SVM-RFE analyses, we implemented the following strategies:

(1) Cross-Validation and Parameter Optimization

LASSO Regression:The optimal regularization parameter (λ, selected via lambda.min) was determined through 10-fold cross-validation (using the glmnet package), minimizing cross-validation error to identify feature genes while avoiding overfitting to training data.

SVM-RFE:Recursive feature elimination was performed with 5-fold cross-validation (implemented via the caret package’s trainControl function: method = "cv", number = 5), iteratively removing low-importance features to optimize classifier performance.

(2) Hyperparameter Selection

LASSO: The regularization parameter (λ) was chosen at the inflection point of the cross-validation error curve.

SVM-RFE: A linear kernel was used, and the penalty parameter (C) was optimized via grid search, with the final parameter combination selected to maximize cross-validation accuracy.

5.Comment.Although protein structure predictions were performed using an advanced tool like AlphaFold, the analyses were presented in a limited manner and their biological significance was not elaborated. Furthermore, the lack of experimental validation of protein-protein interactions limits the impact of this section.

Response:We appreciate the reviewer’s constructive suggestions regarding this section. Below, we provide additional clarifications on the protein structure predictions and outline plans for experimental validation:

(1) Biological Interpretation of Predicted Structures:

AlphaFold-predicted structures of the five core proteins (Amacr, Cyp39a1, Echs1, Osbpl9, and Acsl4) revealed that high-confidence regions are predominantly localized to functional domains, suggesting structural-functional conservation in lipid metabolism. Predicted interactions between “Amacr-Acsl4” and “Echs1-Acsl4” further support their synergistic roles in the mitochondrial β-oxidation pathway, consistent with the GeneMANIA network analysis results.

(2) Planned Experimental Validation

Due to current limitations in sample size and technical resources, experimental validation of protein-protein interactions was not performed in this study. In our next project, we will address this gap through the following approaches:Co-immunoprecipitation (Co-IP): To validate in vivo binding between “Amacr-Acsl4” and “Echs1-Acsl4”ï¼›Fluorescence Resonance Energy Transfer (FRET): To dynamically monitor protein interaction kinetics. These experiments will elucidate the mechanistic roles of these interactions in AP-associated lipid metabolic dysregulation.

6.Comment.While some gene function interpretations in the discussion section are supported by the literature, others appear to be purely speculative. For example, interpretations regarding Mcee or Osbpl9 are not directly supported by experimental data. If such comments are based on probabilities, this should be clearly stated; otherwise, the conclusions may appear overly generalized.

Response:We sincerely appreciate the reviewer's insightful critique regarding the functional interpretations of Osbpl9 and Mcee. As highlighted, the mechanistic studies on these genes in the context of acute pancreatitis remain limited in current literature. Existing publications primarily provide conceptual frameworks (e.g., roles in sterol transport or branched-chain amino acid metabolism) rather than experimental validation of their pathophysiological functions. We acknowledge that our discussion extrapolates preliminary evidence from related fields.To address this gap, future work will prioritize in vitro/in vivo functional validation of Osbpl9 and Mcee using CRISPR/Cas9 models and lipidomic profiling. This multidisciplinary approach will elucidate their precise roles in AP pathogenesis.

7.Comment.The criteria used for evaluating pancreatic histopathology (e.g., Schmidt scoring system)were only briefly mentioned. The specific criteria used to score tissue after H&E staining should be explained in more detail. In particular, the criteria listed in Supplementary Table S1 should be more clearly referenced and described in the main text.

Response:Thank you for your insightful comments. The histopathological scoring criteria for pancreatic tissues were derived from Reference #27: “Schmidt J, Rattner DW, Lewandrowski K, et al. A better model of acute pancreatitis for evaluating therapy. Ann Surg. 1992 Jan;215(1):44-56.” Additionally, we have reorganized the text to ensure that the content of Table S1 is now appropriately referenced and contextualized in the main body of the manuscript to enhance clarity.

8.Comment.There are grammatical errors and phrasing issues in some parts of the manuscript. For instance, spelling errors such as "Notebly" and awkward sentence constructions may affect the readability. A professional level language editing is recommended before publication.

Response:Thank you for your review comments. The relevant content has been revised in the manuscript.

Reviewer 3 Report

Comments and Suggestions for Authors

The manuscript aims to identify genes related to lipid metabolism that can serve a key biomarker for AP using bioinformatics and machine learning approaches. I liked the fact that the authors utilized an in vivo model for validation. I also appreciate the clarity that Figure 1 brings for an overview of the entire study. However, there are some comments that authors can address to improve the manuscript. Point 9 is the most significant question that authors need to address.

  1. What are the real criteria for selection of the database? There are 2695 results for "acute pancreatitis" in the GEO DataSets Database, it is not clear how 3 studies were selected.
  2. Do they use the same or different ways of inducing AP in mice in 3 GEO database? Do they have the same mouse strain and gene background? Please provide more details.
  3. Are there GEO databases based on human AP patients?
  4. The resolution of all figures (except Figure 1) needs to be significantly increased. As they currently stand, they are incredibly difficult, if not impossible, to read.
  5. In Figure 1, I recommend also including the number of genes for AP and lipid metabolism.
  6. Figure 2G: Consider removing Mus musculus (house mouse) in the figure, it is very long compared to pathway name and distracting. A single label on top should suffice.
  7. Section 3.5: Have the author considered using ElasticNet instead of LASSO? Elastic Net is designed to specifically address the multicollinearity issue that LASSO can struggle with. And multicollinearity is very common in gene expression data. It can be done with the same glmnet R package you are already using.
  8. Additionally, in Line 253-255. A strong positive correlation between Osbpl9 and Mcee is another indication that using Elastic Net would be beneficial.
  9. Line 258-261: Which datasets are used for gene validation? It is important to ensure that the validation dataset is distinct from the training dataset to maintain the integrity of the results. The high AUC values of Acsl4 and Mcee are indications of overfitting.  If the authors do not want to validate with extra datasets, consider removing this ROC analysis altogether since later authors validated the genes in vivo.
  10. Consider moving some of the panels in Figure 5 to the supplementary material, it is too crowded right now.
  11. Line 320: 15 total mice or 15 mice for each group?
  12. Line 329: What gene(s) were used as internal control?

Author Response

Reviewer #3:

1.Comment. What are the real criteria for selection of the database? There are 2695 results for "acute pancreatitis" in the GEO DataSets Database, it is not clear how 3 studies were selected.

Response:In selecting datasets for our analysis, we applied the following key criteria:(1) Species Consistency: Prioritized mouse-derived datasets to align with our subsequent AP mouse model validation system, thereby minimizing interspecies variabilityï¼›(2) Group Completeness: Required clear "healthy control" and "acute pancreatitis (AP)" groups with matched sample sizes to avoid group imbalance-induced biasï¼›(3) Data Quality Control: Selected datasets with:Raw data availabilityï¼›Explicit platform information (e.g., GPL339, GPL6246)ï¼›No evident technical batch effects(4) Phenotypic Relevance: Focused on:Pancreatic tissue samplesï¼›AP-specific pathological processes (excluding studies solely addressing complications or other tissues)

Through systematic screening of GEO database using the keyword "acute pancreatitis", we identified three qualifying datasets - GSE3644, GSE65146 and GSE121038. All three datasets met our stringent criteria: â‘ Mouse pancreatic RNA expression profilesâ‘¡Balanced sample sizes (6-9 controls vs. 5-8 AP cases)â‘¢Batch-effect correctable via sva package (post-preprocessing PCA showed good intra-group clustering, Fig. 2B).This comprehensive selection approach ensured the reliability and relevance of our data while maintaining consistency with our experimental model system. The careful consideration of species specificity, group balance, data quality and pathological relevance provided a solid foundation for subsequent analyses.

2.Comment. Do they use the same or different ways of inducing AP in mice in 3 GEO database? Do they have the same mouse strain and gene background? Please provide more details.

Response:All three datasets used the same method of AP induction by rain frogin injection, which may be adjusted according to the different injection time points to eliminate the interference of different AP induction methods on the screening results to the greatest extent. To ensure the coverage of our findings across different genes, we selected a dataset of mice from different genetic backgrounds of the same strain. The strains were all C57BL/6 mice, while GSE3644 performed Mist KO, GSE65146 selected KrasG12D-mutated mice, and BCT KO was performed GSE121038 to compare with litterhouse WT as a control.

3.Comment. Are there GEO databases based on human AP patients?

Response:Yes, there is a database of human patients. A search in the GEO database revealed a dataset of 1826 human patients.This study is exclusively focused on animal-level gene screening and validation, as human sample investigations are encompassed within an ongoing independent study as planned.

4.Comment. The resolution of all figures (except Figure 1) needs to be significantly increased. As they currently stand, they are incredibly difficult, if not impossible, to read.

Response:Thanks, all image resolutions have been increased to 300dpi.

5.Comment. In Figure 1, I recommend also including the number of genes for AP and lipid metabolism.

Response:Thank you for your constructive comments. The relevant modifications have been incorporated into the figures.

6.Comment. Figure 2G: Consider removing Mus musculus (house mouse) in the figure, it is very long compared to pathway name and distracting. A single label on top should suffice.

Response:Thank you for your constructive comments. The relevant modifications have been incorporated into the figures.

7.Comment. Section 3.5: Have the author considered using ElasticNet instead of LASSO? Elastic Net is designed to specifically address the multicollinearity issue that LASSO can struggle with. And multicollinearity is very common in gene expression data. It can be done with the same glmnet R package you are already using.

Response:Thank you very much for your suggestion. We chose LASSO for its superior feature selection in high-dimensional data, which complements SVM-RFE's ability to optimize classification performance. ElasticNet, while addressing multicollinearity, may retain redundant features, reducing interpretability. This combination ensures robust gene selection while maintaining model simplicity and biological relevance.

8.Comment. Additionally, in Line 253-255. A strong positive correlation between Osbpl9 and Mcee is another indication that using Elastic Net would be beneficial.

Response:Thank you so much for your review suggestions! Our reply is the same as Comment 7.

9.Comment. Line 258-261: Which datasets are used for gene validation? It is important to ensure that the validation dataset is distinct from the training dataset to maintain the integrity of the results. The high AUC values of Acsl4 and Mcee are indications of overfitting.  If the authors do not want to validate with extra datasets, consider removing this ROC analysis altogether since later authors validated the genes in vivo.

Response:The current ROC analysis was performed on the training datasets as preliminary validation. A separate clinical validation study using our hospital's acute pancreatitis patient cohort is already planned as the next phase of this research project to further verify these biomarkers' diagnostic performance.

10.Comment. Consider moving some of the panels in Figure 5 to the supplementary material, it is too crowded right now.

Response:Your suggestions are greatly appreciated, but Figure 5 is a GO and KEGG enrichment analysis of a single gene of a signature gene, each of which is of equal importance, and it is currently not possible to screen out parts that can be moved to supplementary material that are not presented in the article. But I adjusted the resolution of the picture to make it easier to watch.

11.Comment. Line 320: 15 total mice or 15 mice for each group?

Response:15 mice for each group,7 of which were used for HE experiments and 8 for RT-qPCR experiments.

12.Comment. Line 329: What gene(s) were used as internal control?

Response: Actb is used as an internal control gene.

Round 2

Reviewer 2 Report

Comments and Suggestions for Authors

Dear Authors,

Thank you for your careful revision. I appreciate the improvements made in response to the initial comments. Specifically:
Abbreviations and terminology are now consistently used.
The introduction clearly states the study aim without including results.
Details regarding cross-validation, train/test splits, and hyperparameter selection in LASSO and SVM-RFE were added, reducing concerns about model overfitting.
Histopathology scoring criteria are now more clearly explained and referenced.
The speculative interpretations of Osbpl9 and Mcee are more cautiously framed.

A few points remain as limitations:
No external human validation was performed. This continues to limit clinical translation, but your acknowledgement is appropriate.
Protein-protein interaction analyses remain predictive only; experimental validation is deferred to future work.

In addition, one important point that still needs improvement:
Code and reproducibility
The actual analysis scripts (R/Python code) are not provided. For bioinformatics and machine learning–based studies, reproducibility is essential. Sharing the scripts—or at least a structured version of the key analysis pipeline—would considerably strengthen the transparency of the study. Many journals now encourage such sharing, and this would make the work more credible and useful for the community.

Overall, the revised manuscript is significantly improved and addresses the majority of the earlier concerns. With the addition of code availability (even partial), the work would reach a level of robustness suitable for publication.

Author Response

Comment 1. The actual analysis scripts (R/Python code) are not provided. For bioinformatics and machine learning–based studies, reproducibility is essential. Sharing the scripts—or at least a structured version of the key analysis pipeline—would considerably strengthen the transparency of the study. Many journals now encourage such sharing, and this would make the work more credible and useful for the community.

Response:Thank you very much for your comments, I have put all the R code I used in the folder of the original data.

Reviewer 3 Report

Comments and Suggestions for Authors

The authors made minimal revisions and did not address the analytical issues. Detailed responses were not added to the manuscript, images remain blurry, and the use of testing data as validation in the ROC analysis is incorrect and should be removed from the manuscript entirely.

  1. I appreciate the response from comment 1 and 2, I do think the details from the response should also be added to the methods, or supplementary methods if the authors prefer.
  2. The figures are still very blurry in the PDF files.
  3. Similarly, answers from comment 11 and 12 should also be reflected in the manuscript.
  4. Comment 9: Again, it is highly inappropriate to run ROC analysis performed on the training datasets as preliminary validation. This part needs to be removed from the manuscript.
  5. Comment 10: My recommendation was to consider transferring either the GO or KEGG analysis to the supplementary materials, rather than selecting individual genes. Figure 5 is still not readable as it stands.

Author Response

Comment 1. I appreciate the response from comment 1 and 2, I do think the details from the response should also be added to the methods, or supplementary methods if the authors prefer.

Response:Thank you for your review comments.I have added this part to the supplementary materials (Table S1-2)

Comment 2. The figures are still very blurry in the PDF files.

Response:Thank you for your review comments.I have adjusted all the images to 600dpi, and I can see them clearly when I open the manuscript to enlarge it.

Comment 3. Similarly, answers from comment 11 and 12 should also be reflected in the manuscript.

Response:Thank you for your review comments.These answers have been reflected in the manuscript.

Comment 4. Comment 9: Again, it is highly inappropriate to run ROC analysis performed on the training datasets as preliminary validation. This part needs to be removed from the manuscript.

Response:Thank you very much for your comments, I have removed this section from the manuscript.

Comment 5. Comment 10: My recommendation was to consider transferring either the GO or KEGG analysis to the supplementary materials, rather than selecting individual genes. Figure 5 is still not readable as it stands.

Response:Thank you very much for your comments, I adjusted the resolution of Figure 5, and now the whole picture has a size of about 300M and a resolution of 600dpi, which can be read clearly. Because this part of the content is the more important and innovative single-gene GSEA enrichment analysis in this article, after careful consideration, I still want to reflect it in the text, thank you very much for your valuable suggestions!

Round 3

Reviewer 2 Report

Comments and Suggestions for Authors

The authors have adequately addressed the concern regarding reproducibility by providing the complete R code alongside the original data. This revision sufficiently enhances transparency and credibility. I am satisfied with the response, and the manuscript can be accepted in its current form.

Reviewer 3 Report

Comments and Suggestions for Authors

Authors have made all the necessary changes to make the manuscript publishable.